# Health Risk Assessment of Antibiotic Pollutants in Large Yellow Croakers from Zhejiang Aquaculture Sites

**DOI:** 10.3390/foods13010031

**Published:** 2023-12-21

**Authors:** Zongjie Li, Yinyin Jin, Xingyu Wang, Liudong Xu, Liyan Teng, Kang Fu, Baoling Li, Yulu Li, Ying Huang, Ning Ma, Feng Cui, Tingting Chai

**Affiliations:** 1College of Food and Health, Zhejiang A & F University, Hangzhou 311300, China; zongjieli2023@163.com (Z.L.); jinyinyin2022@163.com (Y.J.); 17816056411@163.com (X.W.); m19817016211@163.com (L.X.); tly723723@163.com (L.T.); 2Collaborative Innovation Center of Green Pesticide, Zhejiang A & F University, Hangzhou 311300, China; 15936572661@163.com (K.F.); 15847765339@163.com (B.L.); 19106857658@163.com (Y.L.); cuifeng@zafu.edu.cn (F.C.); 3Fishery Resource and Environment Research Center, Chinese Academy of Fishery Sciences, Beijing 100141, China; huangy@cafs.ac.cn (Y.H.); maning@cafs.ac.cn (N.M.)

**Keywords:** quinolones, large yellow croaker, edible parts, health risk

## Abstract

Intensive aquaculture combatting the decline of large yellow croaker populations can trigger bacterial outbreaks, resulting in extensive antibiotic use. In this study, we screened 5 aquaculture sites in the coastal areas of Zhejiang and identified 17 antibiotics in large yellow croakers using UPLC-MS/MS. The distribution and occurrence of antibiotic pollutants were different in the different tissues of large yellow croakers, being primarily dominated by quinolones. Relatively higher average residue levels of enrofloxacin and ciprofloxacin were detected in the inedible parts, specifically the gills (37.29 μg/kg). Meanwhile, relatively high average residue levels of enrofloxacin and ciprofloxacin were also found in the edible parts, particularly in the muscle (23.18 μg/kg). We observed that the residue levels detected in the swim bladder exceeded the prescribed limit for fish muscle, but there is currently no specific regulatory limit established for this particular tissue. Despite the HI values of enrofloxacin and ciprofloxacin being below 0.01, the health risks should not be disregarded. The findings of this research provide significant practical implications for assessing antibiotic contamination and enhancing the risk management of coastal regions.

## 1. Introduction

Aquaculture output continues to experience steady growth annually, whereas the volume of capture-based production has remained relatively stable over the past 20–30 years [1]. By 2030, global aquaculture production is projected to reach 103 million metric tons, surpassing the capture fisheries sector by 6 million metric tons [2]. Aquaculture possesses the capability to produce large quantities of products within confined spaces, overcoming the constraints associated with capturing wild species [3]. As the global population continues to grow, there is a rising need for protein to meet the increasing demand [4]. Aquaculture, one of the fastest-growing sectors within animal production, has become a pivotal global industry in meeting the rising demand for edible protein and nutrients sourced from seafood, thus serving as a crucial sector in global food production [5]. Marine aquaculture plays a crucial role in economic development, with approximately 74% of its production coming from fish farming [6]. China, as the world’s largest supplier of marine aquaculture products, contributes about 60% of global production [7,8]. The large yellow croaker (*Larimichthys crocea*) is one of the most significant marine economic fish being extensively cultivated along the southeastern coast of China [9]. Yellow croaker, known for its delectable taste, tender texture, and richness of nutrients, is highly regarded in China for its significant commercial and economic value [10]. According to the China Fisheries Statistical Yearbook, the annual production of large yellow croakers surpassed 254,224 tons in 2021. This accounted for 13.79% of China’s total mariculture yield [11]. Zhejiang province, being a maritime giant in China with vast coastal areas and abundant marine resources, serves as a vital region for marine aquaculture [12], especially the primary farming regions for the large yellow croaker [11].

The swift transition to intensive aquaculture has caused a decline in water quality within aquafarms, resulting in a notable increase in bacterial diseases [13]. Antibiotics combat bacteria, fungi, and parasites and are grouped by mechanisms, such as sulfonamides (SAs), macrolides (MLs), tetracyclines (TLs), and quinolones (QNs) [14,15]. To prevent or treat infections in aquaculture, a significant quantity of antibiotics is employed through methods including oral administration, immersion baths, pond spraying, and injections [16,17]. The global consumption of antibiotics compounds in aquacultural industries exceeded 10 Mt. in 2017 and will increase by 33% by 2030 [18]. Based on the World Health Organization’s report, Asia consumes 70% of the world’s antibiotics, with China accounting for 70% of this consumption [15]. The extensive use of antibiotics inevitably leads to substantial levels of antibiotic residues in aquaculture environments. Many countries have enacted varied policies aimed at reducing the widespread use of antibiotics in aquaculture products. For example, the European Union is recognized for implementing stringent regulations on the maximum permissible limits of antibiotic residues in animal food products [19]. In Asian aquaculture, due to a lack of institutional regulation, farmers often excessively use or misuse antibiotics [20], especially in China, leading to the emergence of certain issues.

Recently, antibiotics have garnered global attention due to their widespread usage and potential adverse impacts on both the ecosystem and human health [21]. While antibiotics effectively treat fish diseases and boost aquatic product yields, intensive research reveals that only a small fraction of these antibiotics undergoes breakdown, with the majority either remaining within the tissues or being excreted into the external environment [22]. Han et al. found enrofloxacin and ciprofloxacin in turbots from a marine aquaculture farm near the Yellow Sea in North China, with concentrations of 24.75 μg/kg and 9.62 μg/kg, respectively [21]. Li et al.’s study documented the presence of antibiotics in 10 common freshwater aquaculture products in eastern China, revealing that enrofloxacin was detected in the muscle samples at a high rate of 85%, with the highest detected concentration reaching 18 μg/kg [23]. These studies unveil the widespread presence of antibiotic residues in aquaculture products. The consumption of fish treated with antibiotics may have negative effects on human health [24], such as impacting the development of children’s teeth and potentially inducing allergic reactions [7]. Additionally, exposure to antibiotics may promote the development of antibiotic-resistant bacteria and genes [25,26], posing a serious threat to the health of both animals and humans [27]. Therefore, regulating the use of antibiotics in seafood culture and encouraging people to choose safe and reliable food sources is crucial for maintaining human health. Several past studies have assessed the possible health risks linked to consuming fish that have been exposed to antibiotics. The consumption of fish in Canada [28] and Egypt [29] demonstrated minimal negative effects on consumers. In contrast, a study in China [21] found significant risks associated with consuming fish.

Our study aims to investigate the distribution of antibiotics in yellow croakers, detecting 42 commonly used antibiotics in the marine food chain, including four classes: TLs, SAs, QNs, and MLs. Furthermore, we utilized antibiotic concentrations to assess the issues related to the human health risks associated with consuming yellow croakers from aquaculture farms along the coast of Zhejiang province. This study provides crucial insights into antibiotic distribution patterns in the different parts of yellow croakers from aquaculture farms in Zhejiang province, offering essential guidance for antibiotic risk assessments.

## 2. Materials and Methods

### 2.1. Sample Collection

Five mariculture sites were selected in the Zhejiang province: Site 1 (121.13° E, 28.25° N), Site 2 (121.86° E, 28.46° N), Site 3 (121.76° E, 29.53° N), Site 4 (122.60° E, 30.81° N), and Site 5 (122.34° E, 29.92° N). The locations of the sampling sites are shown in Figure 1. In each site, three parallel samples were collected and swiftly transported back to the laboratory within 2 h for immediate dissection and sampling.

### 2.2. Chemicals and Supplies

Antibiotic standards for this study were sourced from Shanghai Yuanye Bio-Technology Co., Ltd. in Shanghai, China. The investigation focused on four categories of antibiotics within nine distinct sections of the yellow croaker fish. These categories included QNs, SAs, MLs, and TLs, comprising a total of 42 different compounds, as listed in Table 1. All standard stock solutions were stored in opaque containers at a temperature of −20 °C, with fresh stock solutions being prepared anew every three months. The solvent used was methanol of HPLC-gradient grade, procured from Merck KGaA in Darmstadt, Germany. Ultra-pure water was generated by utilizing the Milli-Q Integral Water Purification System produced in Billerica, MA, USA. Ethylenediaminetetraacetic acid disodium salt (Na2EDTA), of HPLC-gradient grade and with a minimum purity of 99.0%, formic acid of HPLC-gradient grade and with a minimum purity of 98%, and ammonia solution of chromatography grade with a minimum purity of 25% were all acquired from Aladdin in Shanghai, China.

### 2.3. Fish Samples Extraction and Analysis

During the dissection process, various tissues, including the brain, muscle, swim bladder, gonads, heart, liver, kidney, intestine, and gills, were extracted from the yellow croaker. These tissue samples were utilized for subsequent analysis and experiments. All samples were stored at −20 °C in the laboratory for further analysis. Referring to the extraction methods outlined in previous studies [30,31], we conducted the optimization of antibiotic extraction from various samples of yellow croaker tissues, weighing 0.5 g each. Sequentially, 50 mg of Na2EDTA, 0.5 mL of phosphate-buffered saline (pH = 4), and 0.5 mL of acetonitrile were successively introduced. The resulting mixture underwent vortexing, followed by extraction and subsequent centrifugation to isolate the supernatant. This supernatant, containing the target analytes, was meticulously transferred to a 2 mL centrifuge tube. In the subsequent step, 0.5 mL of phosphate-buffered saline (pH = 8) and 0.5 mL of acetonitrile were sequentially added to the tube. The resulting mixture underwent vigorous vortexing, extraction, and centrifugation. The resulting supernatants were combined and thoroughly mixed. To eliminate interfering substances, 40 mg of purification materials (20 mg PSA and 20 mg C18) was added. The mixture was vortexed for 10 s and was then centrifuged at 8000 rpm for 3 min. The clarified supernatant was further purified by passing it through a 0.22-μm organic filter membrane and stored in amber vials at −20 °C for subsequent analysis.

### 2.4. Instrumental Conditions

UPLC-MS/MS analysis was performed using a UPLC system (LC-20A, Shimadzu, Kyoto, Japan), coupled with a triple quadrupole mass spectrometer (Shimadzu LC-MS 8050, Kyoto, Japan). Chromatographic separation was achieved on a Waters ACQUITY BEH C18 column (2.1 mm× 50 mm; 1.7 μm; Milford, MA, USA) with gradient elution, at 40 °C. Eluent A was of 0.1% *v*/*v* formic acid in water, while eluent B was acetonitrile. The eluent gradient was 0 min with 5% B, 0.5 min with 15% B, 1.5 min with 75% B, 3 min with 95% B, 5 min with 95% B, and 6 min with 5% B, using a mobile flow rate of 0.3 mL/min. The injection volume was 1 μL.

MS/MS detection was performed in the positive ionization mode (ESI+) using multiple reaction monitoring (MRM). The MS parameters were configured with a nebulizer gas flow of 3 L/min, a heater gas flow of 10 L/min, an ionization voltage of +3500 V, and with the interface, transfer line, and heater block temperatures set at 300 °C, 250 °C, and 400 °C, respectively. The drying gas flow rate was maintained at 10 L/min. The ion information is presented in Table 2.

### 2.5. Quality Assurance and Quality Control

Standard solutions containing different antibiotics at concentrations of 1 µg/L, 5 µg/L, 10 µg/L, 50 µg/L, 200 µg/L, and 500 µg/L were prepared and subjected to analysis using UPLC-MS/MS.

To enhance the experimental accuracy, researchers often identify and rectify systematic errors and mitigate the interference caused by the sample matrix effect through labeling recovery experiments. In this study, 42 types of antibiotics were mixed and then added to blank large yellow croaker meat samples in the recovery experiment. Blank samples were spiked with target antibiotics at three concentration levels (1, 10, and 50 μg/kg), with five replicates to confirm the recovery percentages. Statistical analysis was performed using the standard curve. Following UPLC-MS/MS analysis, a linear relationship between the concentration and response peak area was observed, with an R^2^ value exceeding 0.99, indicating a strong linear correlation. The limits of detection (LODs) and limits of quantification (LOQs) were established, following the methodology outlined in Shaaban et al.’s research [32]; they were established using signal-to-noise ratios of 3:1 and 10:1, respectively. The recovery rates for QNs in the tissues varied from 60.2% to 108%, with those of MLs from 64.6% to 119%, TLs from 63.3% to 86%, and SAs from 61.4% to 104%. The detailed results are shown in Table 3. All the above results indicated that the method used exhibited strong linear relationships and demonstrated exceptional sensitivity.

### 2.6. Health Risk Assessment

Research was carried out to evaluate the potential health hazards to humans resulting from the consumption of fish tainted with antibiotics. The estimated daily intake (EDI) of antibiotics was determined by considering the quantity of antibiotic residues present in the fish. The calculation followed the formula: EDI=(C×FR)/BW

In this equation, C denotes the antibiotic content in fish (measured in μg/g), FR stands for the estimated daily fish consumption by Chinese adults (49.30 g/day) [33], and BW represents the assumed average body weight of adults (in kilograms, calculated at 60 kg).

A hazard quotient (HQ) serves as a key indicator for evaluating the potential risk of a chemical or environmental pollutant to human health. The acceptable daily intake level (ADI) defines the safe daily dose of a chemical to which humans can be exposed chronically without adverse health effects, typically expressed in milligrams per kilogram of body weight per day (mg/kg bw/day). It signifies the quantity of a particular chemical that humans can safely encounter daily without harming their health. ADI calculations find widespread application in fields such as toxicology research, food safety evaluation, and environmental risk assessment. To gauge the health risks, this study employed an HQ, which was calculated by comparing the EDI of the chemical with the corresponding ADI. The calculation followed the formula:HQ=EDI/ADI

Since there are multiple antibiotic residues in fish samples, it is necessary to conduct a cumulative risk assessment through the calculation of a hazard index (HI). This value is obtained by summing the HQ value of each identified analyte within the various antibiotic classes under examination. If the HI was less than 1.0, the health risk was deemed negligible; however, if the HI equaled or exceeded 1.0, the health risk was considered significant. Enrofloxacin and ciprofloxacin are the primary quinolone antibiotics with the highest detection rates and concentrations in this study. Due to their significantly higher detection rates and concentrations compared to other quinolone antibiotics, a dedicated risk assessment was conducted for these specific compounds.

### 2.7. Statistical Analyses

We used ArcMap 10.8 to process the geographic information and other graphing processes were conducted using GraphPad Prism 8. 

## 3. Results and Discussion

### 3.1. QNs as the Principal Antibiotic in Large Yellow Croakers from Coastal Aquacultures

We evaluated the distribution of differential types of antibiotics in large yellow croaker from 5 coastal aquacultures, and 17 out of the 42 target antibiotics were detected (Figure 2 and Table 4). The detected antibiotics belonged to the QN, TL, ML, and SA groups. In Site 1, we found 16 antibiotics in its large yellow croakers. QNs accounted for 88.9% of the detected antibiotics in these large yellow croakers, ranging from 0.03 to 21.30 μg/kg. The proportions of TLs, MLs, and SAs were 22.2%, 11.1%, and 11.1%, respectively. For the QNs, garenoxacin showed its highest concentration (21.30 μg/kg) in the fish gills. In Site 2, 15 antibiotics were detected in its large yellow croakers. QNs constituted 88.9% of the antibiotics detected in its large yellow croakers, ranging from 0.64 to 11.03 μg/kg. The TLs, MLs, and SAs accounted for 11.2%, 22.2%, and 33.3% of the total, respectively. In the fish livers, ciprofloxacin exhibited the highest levels (13.26 μg/kg) among the QNs. In Site 3, 11 antibiotics were discovered in its large yellow croakers. QNs accounted for 100% of the antibiotics identified in its large yellow croakers, ranging from 0.26 to 87.22 μg/kg. The TLs, MLs, and SAs made up 22.22%, 22.22%, and 11.11% of the total, respectively. Enrofloxacin displayed the highest concentrations (87.22 μg/kg) among QNs in the swim bladders. In Site 4, a total of 10 different antibiotics were detected in its large yellow croakers. QNs constituted 100% of the antibiotics found in its large yellow croakers, with concentrations ranging from 0.69 to 44.14 μg/kg. The TLs accounted for 11.1% of the total, while MLs constituted 22.2%, and SAs made up 44.4%. Among the QNs, enrofloxacin showed the highest concentrations (44.14 μg/kg) in the fish intestines. In Site 5, we detected 15 antibiotics in its large yellow croakers. QNs accounted for 100% of the antibiotics detected in its large yellow croakers, with concentrations ranging from 0.24 to 33.41 μg/kg. The proportions of TLs, MLs, and SAs were all at 22.2%. Among the QNs, enrofloxacin reached the highest concentrations (33.41 μg/kg) in fish gills.

QNs are widely utilized in veterinary medicine and aquaculture to prevent and treat diseases in animals [34]. For instance, a previous study on antibiotic use in coastal aquaculture within areas like Ningbo, Wenzhou, and Taizhou discovered the presence of quinolone antibiotics being incorporated into aquaculture feed, with levels reaching as high as 140 milligrams per kilogram [35]. It is widely recognized that QNs (enrofloxacin, norfloxacin, and ofloxacin) exhibit low water solubility, and, as a consequence, tend to accumulate in aquatic environments and products [36]. Shaaban et al. surveyed various antibiotics in fish consumed in the Saudi market, finding QNs with a highest detection rate of 92.5% and a maximal detected concentration of 121.10 μg/kg [33]. Additionally, He et al.’s research revealed that eels exhibited the highest detection concentrations of QNs at 185.7 ± 19.9 μg/kg [37], while Li and colleagues found a 100% detection rate of QNs in shrimp during their study [38]. These findings align with our own research results, wherein QNs dominate in terms of both concentration and detection rate among the five regions studied. Our results showed that the frequent detection of 10 QNs highlighted in large yellow croakers is possibly due to their extensive use in this area and their stability in edible animal tissues [37]. Therefore, exploring the potential impact of QNs in large yellow croakers on human health is crucial, emphasizing the urgent need to regulate the use of QNs in aquaculture to prevent the risk of excessive antibiotic intake.

### 3.2. Enrofloxacin and Ciprofloxacin with Higher Frequency in Differential Tissues

As illustrated in Figure 3, the distribution of detected QNs was analyzed in edible parts and inedible parts. In terms of inedible tissues, 6, 4, 3, 3, 2, and 2 out of 10 QNs were detected in the heart, kidney, gonads, gills, intestine, and liver, respectively. In the gills and hearts, garenoxacin exhibited maximal concentrations of 15.19 μg/kg and 4.89 μg/kg, respectively. Enrofloxacin levels were found to have the second-highest levels (8.62 μg/kg) in the gills. The detection frequency of samples was evaluated by combining the presence of ciprofloxacin and enrofloxacin, with ciprofloxacin being the primary metabolite of enrofloxacin. Enrofloxacin was observed at the maximal levels in the gonads (11.92 μg/kg), kidneys (8.21 μg/kg), and intestine (22.47 μg/kg), respectively. Ciprofloxacin levels were found to be the second highest, following enrofloxacin, in the gonads (3.89 μg/kg), kidneys (8.18 μg/kg), and intestine (9.35 μg/kg), respectively. Ciprofloxacin, as the metabolite of enrofloxacin, was detected and exhibited the highest concentration (6.54 μg/kg) in the liver. For the edible tissues, 4, 4, and 2 out of 10 QNs were detected in the muscle, swim bladder, and brain, respectively. Enrofloxacin was observed at the maximal levels in the swim bladder (17.63 μg/kg) and brain (7.14 μg/kg), respectively. Ciprofloxacin was also detected at the second-most maximal levels in the swim bladder (16.42 μg/kg) and brain (6.01 μg/kg), respectively. In the muscle sample, ciprofloxacin was observed at the maximal concentration of 9.04 μg/kg, and enrofloxacin was found to be at the second-highest level (6.98 μg/kg).

According to our results, the average residues and detection rates of enrofloxacin and ciprofloxacin among the QNs were relatively high in large yellow croakers. Recent studies have shown that enrofloxacin and ciprofloxacin residues are of significant concern [21]. Enrofloxacin is widely used due to its broad spectrum of antibacterial activity, strong bactericidal effects, and rapid action [39], making it one of the most frequently detected veterinary antibiotics in aquatic products [40]. Griboff et al., during their research on fish in the Argentine market, found enrofloxacin to be present in 100% of their samples [41]. In a study conducted by Wang et al., 160 cultured fish samples from Shandong Province were investigated, revealing that enrofloxacin exhibited the highest residue levels of up to 260 μg/kg [42]. Li et al. found that in their study of antibiotic residues in aquaculture regions in eastern China, including Tai Zhou, enrofloxacin was observed, with a detection rate as high as 85% [23]. This corresponds with our research findings, emphasizing the widespread use of enrofloxacin in cultured fish. Enrofloxacin has received authorization for use, while ciprofloxacin has been prohibited in Chinese aquaculture for approximately two decades. In research conducted by Song et al. on the primary aquaculture provinces in China, it was observed that ciprofloxacin exhibited relatively high detection concentrations (23.92 μg/kg) in the Chinese mitten crab (*Eriocheir sinensis*) [43]. Liu et al. found up to 31 μg/kg of ciprofloxacin in freshwater fish from China’s northern coastal aquaculture [44]. Likewise, our findings revealed relatively high levels of ciprofloxacin in large yellow croakers. The occurrence of ciprofloxacin in these cultured fish can be explained by considering that its presence is primarily influenced by the degradation of enrofloxacin [43]. Therefore, these two substances are frequently detected together in aquatic products, and they are often collectively considered in the calculation of maximum residue limits. The residues of enrofloxacin and ciprofloxacin in aquatic products can be transmitted to humans through the food chain, potentially causing adverse effects. Regarding this issue, it is essential to closely monitor the frequent detection of enrofloxacin and its metabolite, ciprofloxacin.

### 3.3. Tissue Distribution of Enrofloxacin Residue in Edible and Inedible Parts

The total concentrations of enrofloxacin and ciprofloxacin in the edible or inedible parts of large yellow croakers from different coastal aquaculture sites in Zhejiang were calculated and the results are presented in Figure 4. In Site 1, the maximal residue limit of enrofloxacin in the muscle (11.70 μg/kg) and brain (10.84 μg/kg) accounted for a larger percentage in the edible parts, e.g., 20.6% in the muscle and 19.1% in the brain. Within the confines of the inedible parts, the maximal residue limit of enrofloxacin in the liver (12.16 μg/kg) and gills (10.77 μg/kg) accounted for the largest percentage, specifically, 21.4% in the liver and 19.1% in the gills. In Site 2, the highest residue limit of enrofloxacin in the muscle (16.28 μg/kg) and brain (14.83 μg/kg) represents a significant proportion in the edible parts, e.g., 15.8% in the muscle and 14.4% in the brain. Among the inedible parts, the highest residue limit of enrofloxacin in the kidneys (14.92 μg/kg) and liver (13.26 μg/kg) constitutes a significant proportion, specifically, 14.5% in the kidneys and 12.8% in the liver. Among the 5 coastal aquacultures analyzed, in Site 3, the maximal residue was detected in all edible parts. Among the edible parts, the maximal residue limit of enrofloxacin in the swim bladder (147.81 μg/kg) and muscle (38.33 μg/kg) accounted for the largest percentage, e.g., 32.4% in the swim bladder, and 8.4% in the muscle. Among the inedible parts, the maximal residue limit of enrofloxacin in the gills (95.32 μg/kg) and intestine (72.56 μg/kg) constituted a significant percentage, e.g., 20.9% in the gills, and 15.9% in the intestine. In Site 4, the highest residue limit of enrofloxacin in the swim bladder (28.21 μg/kg) and muscle (21.81 μg/kg) represented a significant proportion in the edible parts, e.g., 12.9% in the swim bladder and 10.0% in the muscle. Among the inedible parts, the highest residue limit of enrofloxacin in the intestine (51.98 μg/kg) and gills (30.83 μg/kg) constituted a significant proportion, specifically, 23.7% in the intestine and 14.1% in the gills. In Site 5, the maximal residue limit of enrofloxacin in the muscle (27.77 μg/kg) and brain (20.28 μg/kg) accounted for a substantial proportion in the edible parts, with muscle representing 15.1% and the brain representing 11.0%. Among the inedible parts, the highest residue limit of enrofloxacin in the gills (40.25 μg/kg) and kidneys (31.26 μg/kg) accounted for a substantial proportion, specifically, 21.9% in the gills and 17.0% in the kidneys.

Enrofloxacin and ciprofloxacin exhibited variations in their distribution across the different tissues. In the inedible parts of the large yellow croaker, specifically in the liver samples, we only detected the presence of ciprofloxacin residues and did not find any traces of enrofloxacin. This phenomenon can be attributed to the liver being the primary metabolic organ within the organism. Enrofloxacin is likely metabolized entirely in the liver, converting it into its metabolite, ciprofloxacin. Previous studies have indicated that enrofloxacin primarily undergoes metabolic processes in the liver [45]. Shan et al. conducted research on the tissue-residue profiles of enrofloxacin in crucian carp, discovering that ciprofloxacin was most commonly distributed in the liver [46]. Consequently, our study only identified the residues of ciprofloxacin in the liver samples of the large yellow croaker, providing valuable insights into the metabolic pathways of enrofloxacin in biological tissue. Furthermore, we found relatively high levels of antibiotic residues in fish gills, especially in the samples from Site 3 and Site 5, where the inedible parts exhibited the highest proportion of antibiotic residues. In Zhang et al.’s study on the tissue-specific bioaccumulation of antibiotics in marine aquaculture organisms, it was found that enrofloxacin also exhibited relatively high detection concentrations in fish gills, ranging from 0.77 to 126 μg/kg [47]. Hua et al. indicated that antibiotics can be absorbed through the skin and gills of fish from the water, accumulating in fish tissues [48]. This finding indicated that gills serve as the primary route for antibiotics to enter the fish body, leading to the observed high levels of antibiotic residues in fish gills due to direct exposure. Recent studies have shown that QNs tend to accumulate more readily in fish muscle [49]. Although, within the edible parts, the antibiotic concentration in muscle was not the highest recorded, except in fish from Site 1 and Site 2, this concentration played a crucial role in the total body burden of enrofloxacin, which was due to the fact that muscle accounted for approximately 67.0% of the fish’s total mass [50]. Considering that fish muscle constitutes the primary component of the human diet, this could pose a significant potential source of enrofloxacin for human exposure. Additionally, the residue levels in the swim bladder surpassed the limit of 100 μg/kg that has been specified for muscle tissue in China [51] at Site 3. The swim bladders are rich in collagen and are widely used in food and medicine, making this finding a cause for major concern and requiring alerts [52]. However, due to the lack of explicit regulations on maximum residue limits, there is an urgent need to conduct a risk assessment for swim-bladder residue levels and establish the corresponding standards. Regarding this matter, attention should be paid to the presence of residues in the edible parts.

### 3.4. Evaluation of the Enrofloxacin and Ciprofloxacin Risk to Human Health

The large yellow croakers examined in this research were classified as fit for human consumption. Utilizing a previously published assessment method, we calculated the daily intake of antibiotics in large yellow croakers, considering a worst-case scenario in China by using the maximal concentrations in the calculations. In 5 aquaculture sites, the EDIs of QNs ranged from 3.82 × 10^−3^ ng/kg bw/d for enrofloxacin to 18.75 × 10^−3^ ng/kg bw/d for ciprofloxacin. Using the EDI values and the corresponding ADIs, hazard quotients (HQ) for individual antibiotics were computed (as presented in Table 5 and Figure 5) to assess the human health risks stemming from dietary exposure to antibiotics. In Site 1, the HQ values of enrofloxacin and ciprofloxacin were 9.34 × 10^−4^ and 6.17 × 10^−4^, respectively. In Site 2, the HQ values for enrofloxacin and ciprofloxacin were 1.22 × 10^−4^ and 9.37 × 10^−4^, respectively. Site 3 had HQ values of 3.02 × 10^−3^ for enrofloxacin and 2.06 × 10^−3^ for ciprofloxacin. In Site 4, the HQ values for enrofloxacin and ciprofloxacin were 1.46 × 10^−3^ and 1.43 × 10^−3^, respectively. Site 5 recorded HQ values of 1.28 × 10^−3^ for enrofloxacin and 2.40 × 10^−3^ for ciprofloxacin. The HQs of enrofloxacin and ciprofloxacin for all the 5 aquaculture areas analyzed were lower than 1.0. By considering a similar toxicological mode of action for substances within the same class, we calculated hazard index values (HI = ΣHQ) to assess the potential human health risk associated with consuming large yellow croakers from this area [53]. In Site 1, the HI value was calculated at 1.60 × 10^−3^, while Site 2 had an HI value of 2.17 × 10^−3^. All of them were relatively low. The HI value was 5.09 × 10^−3^ in Site 3, which had the highest HI value among the five sites. In Site 4, the HI value was 2.89 × 10^−3^, and Site 5 reported an HI value of 3.68 × 10^−3^. The calculated HI values of ΣQNs consistently remained below 1.0, indicating that the potential adverse effects from consuming these large yellow croakers were relatively low.

While our study indicates a low health risk linked to consuming large yellow croakers, the controversial worldwide use of enrofloxacin renders the health risks associated with ingesting enrofloxacin and ciprofloxacin in this way non-negligible. Enrofloxacin has been banned in aquaculture in countries like the US, Canada, and Chile due to the risks it poses to consumers [54,55]. Nevertheless, enrofloxacin is still being employed in Chinese aquaculture [40]. Wang et al.’s assessments in Zhejiang province unveiled the widespread utilization of enrofloxacin in aquaculture, encompassing its application in the cultivation of large yellow croakers [56]. It is estimated that enrofloxacin concentrations are likely to rise, leading to increased HQ values and higher risks of human exposure through fish consumption [57]. Furthermore, recent research indicates that chronic toxic effects as a result of prolonged low-level exposure to antibiotics have already been directly or indirectly proven [58]. For example, exposure to environmental concentrations of enrofloxacin has been observed to induce immune system impairments in yellow catfish [59], while ciprofloxacin exposure has been linked to an imbalance in the oxidative defense system of crucian carp [60]. Ren et al.’s study indicates that the combined toxicity of enrofloxacin and ciprofloxacin in aquatic environments exhibits a synergistic effect [61]. This previous study has overlooked the need to assess the interaction and combined toxicity of enrofloxacin and ciprofloxacin. Enrofloxacin and ciprofloxacin residues have consistently been found concurrently in large yellow croakers. Therefore, further investigation is necessary to assess the human health risks associated with mixtures of enrofloxacin and ciprofloxacin. Our findings emphasize the risks that these drug combinations pose to human health, providing new perspectives for implementing better risk assessment and management strategies, especially concerning large yellow croakers.

## 4. Conclusions

We conducted tests on 9 different body parts of large yellow croaker samples from five separate aquaculture sites and found a total of 17 antibiotics in the samples. Among them, quinolones were one of the four most frequently detected types. Among the 10 quinolone antibiotics, enrofloxacin and ciprofloxacin had the highest detection rates and residual concentrations, indicating that large yellow croakers in coastal farms in Zhejiang Province are primarily contaminated by enrofloxacin. In the inedible parts, enrofloxacin and ciprofloxacin primarily exhibited higher residues in the gills, with only the metabolite ciprofloxacin being detected in the liver. In the edible parts, muscle was the tissue where the enrofloxacin and ciprofloxacin residues were comparatively higher. The residue levels in the swim bladder of fish from Site 3 surpassed the maximal residue limit for muscle. Considering that swim bladders are commonly used as medicine and food in southern China, this situation requires significant attention. Although the HI values are all less than 1.0, the long-term low-dose consumption of enrofloxacin and ciprofloxacin can still pose risks to human health. Therefore, the health risks of QNs associated with large yellow croakers from Zhejiang’s coastal farms were non-negligible. These findings offer insights into the assessment of human health risks associated with consuming fishery products. Additionally, it is vital to note that the limited sample size in this study may not fully represent the health risks associated with consuming aquatic products from this region, necessitating a larger-scale monitoring program.

## Figures and Tables

**Figure 1 foods-13-00031-f001:**
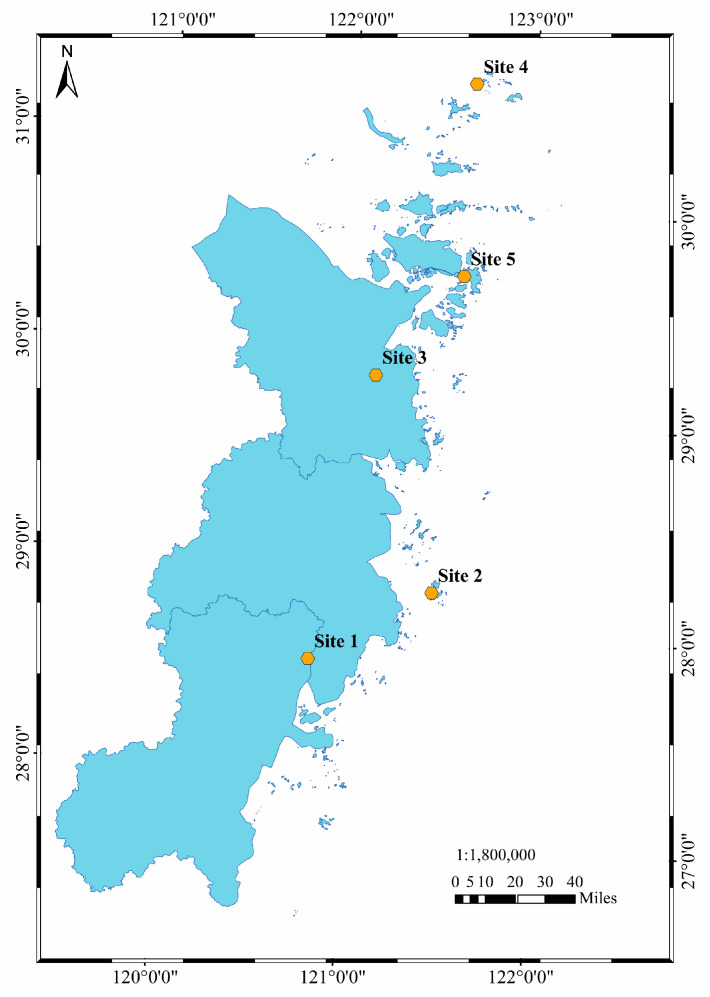
Locations of the sampling sites in Zhejiang province.

**Figure 2 foods-13-00031-f002:**
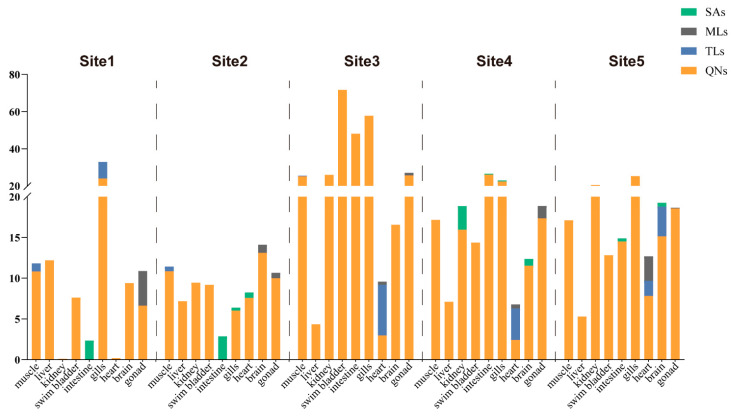
Stacked plot of antibiotics in large yellow croakers from the five sampling sites within the Zhejiang aquaculture industry.

**Figure 3 foods-13-00031-f003:**
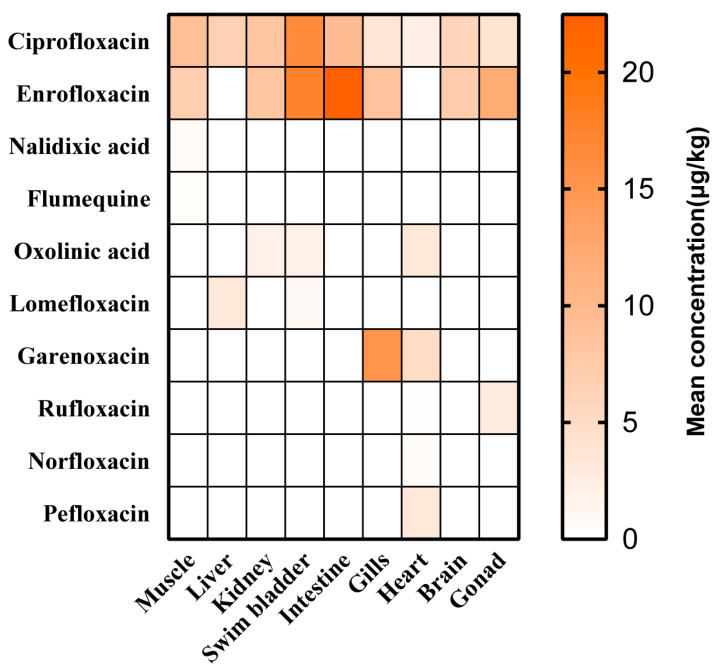
Mean concentration of QNs in the different tissues.

**Figure 4 foods-13-00031-f004:**
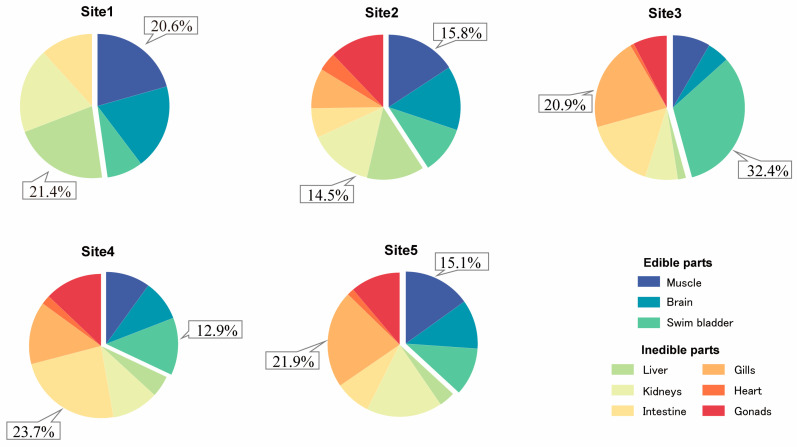
Proportions of the highest enrofloxacin and ciprofloxacin residues in different large yellow croaker tissue samples at the five sites.

**Figure 5 foods-13-00031-f005:**
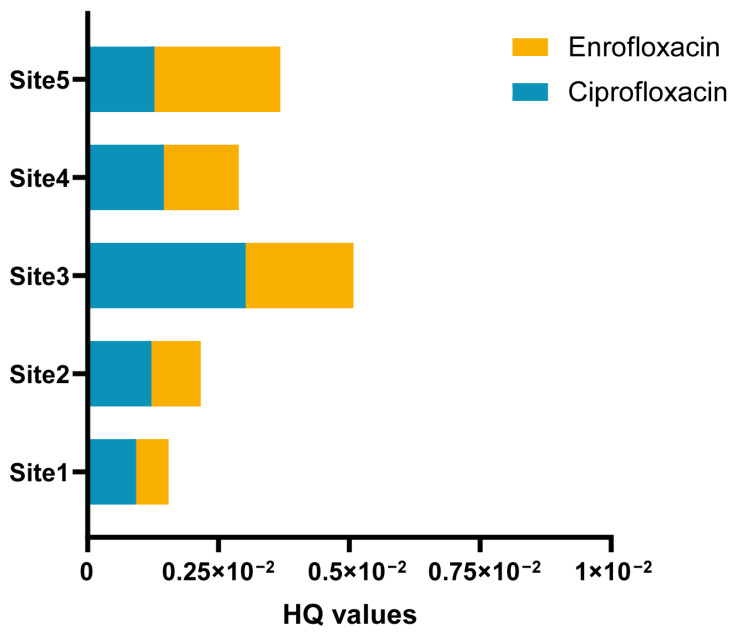
The hazard quotients (HQ) of enrofloxacin and ciprofloxacin for human consumption under the worst-case scenarios.

**Table 1 foods-13-00031-t001:** Detailed information on the 42 compounds.

NO.	Category	Compound	Acronym	CAS	Purity
1	QNs	Cinoxacin	CIN	28,657-80-9	98%
2	QNs	Ciprofloxacin	CIP	85,721-33-1	98%
3	QNs	Danofloxacin	DAN	112,398-08-0	98%
4	QNs	Difloxacin	DIF	98,106-17-3	98%
5	QNs	Enoxacin	ENO	74,011-58-8	98%
6	QNs	Enrofloxacin	ENR	93,106-60-6	98%
7	QNs	Fleroxacin	FRX	79,660-72-3	98%
8	QNs	Flumequine	FMQ	42,835-25-6	98%
9	QNs	Garenoxacin	GRX	194,804-75-6	99%
10	QNs	Gatifloxacin	GTFX	112,811-59-3	99%
11	QNs	Grepafloxacin	GPFX	119,914-60-2	98%
12	QNs	Lomefloxacin	LOM	98,079-51-7	98%
13	QNs	Marbofloxacin	MAR	115,550-35-1	99%
14	QNs	Moxifloxacin	MOX	151,096-09-2	98%
15	QNs	Nadifloxacin	NDFX	124,858-35-1	98%
16	QNs	Nalidixic acid	NALA	389-08-2	98%
17	QNs	Norfloxacin	NRFX	70,458-96-7	98%
18	QNs	Ofloxacin	OFL	82,419-36-1	99%
19	QNs	Orbifloxacin	ORB	113,617-63-3	98%
20	QNs	Oxolinic acid	OXOA	14,698-29-4	98%
21	QNs	Pefloxacin	PEFX	70,458-92-3	98%
22	QNs	Pipemidic acid	PIPE	51,940-44-4	98%
23	QNs	Piromidic acid	PIRO	19,562-30-2	98%
24	QNs	Rufloxacin	RUF	101,363-10-4	98%
25	QNs	Sitafloxacin	SIT	127,254-12-0	98.5%
26	QNs	Sparfloxacin	SPA	110,871-86-8	99%
27	QNs	Tosufloxacin	TFLX	108,138-46-1	98%
28	QNs	Trovafloxacin	TVX	147,059-72-1	98%
29	SAs	Sulfadiazine	SDZ	68-35-9	98%
30	SAs	Sulfamethazine	SMT	57-68-1	98%
31	SAs	Sulfamethoxazole	SMX	723-46-6	98%
32	SAs	Sulfamethoxypyridazine	SMPD	80-35-3	98%
33	SAs	Sulfapyridine	SPY	144-83-2	98%
34	SAs	Sulfaquinoxaline	SQX	59-40-5	97%
35	SAs	Trimethoprim	TMP	738-70-5	98%
36	MLs	Clarithromycin	CLA	81,103–11-9	98%
37	MLs	Erythromycin	ERY	114-07-8	98%
38	MLs	Roxithromycin	ROX	80,214-83-1	98%
39	MLs	Tylosin	TYL	1401-69-0	93%
40	TLs	Chlortetracycline	CTC	64-72-2	90%
41	TLs	Oxytetracycline	OTC	2058-46-0	98%
42	TLs	Tetracycline	TC	60-54-8	95%

**Table 2 foods-13-00031-t002:** Optimized MS/MS parameters for the emerging contaminants.

Category	Compound	MRM1	Q1 Pre	CE1	Q3 Pre	MRM2	Q1 Pre	CE2	Q3 Pre
			Bias/V		Bias/V		Bias/V		Bias/V
QNs	Cinoxacin	263.1 > 245.1 *	−16	−15	−16	263.1 > 217.1	−16	−21	−14
QNs	Ciprofloxacin	332.1 > 288.1 *	−17	−18	−20	332.1 > 314.3	−17	−22	−21
QNs	Danofloxacin	358.2 > 340.1 *	−12	−23	−23	358.2 > 82.1	−13	−44	−14
QNs	Difloxacin	400.4 > 356.5 *	−12	−18	−12	400.4 > 299.4	−12	−28	−20
QNs	Enoxacin	321.3 > 303.4 *	−12	−20	−20	321.3 > 234	−12	−23	−15
QNs	Enrofloxacin	360.2 > 316 *	−12	−19	−21	360.2 > 245	−12	−26	−16
QNs	Fleroxacin	370.1 > 326.1 *	−11	−20	−22	370.1 > 269.2	−11	−25	−18
QNs	Flumequine	262.1 > 244.1 *	−15	−19	−16	262.1 > 202	−15	−32	−13
QNs	Garenoxacin	427.4 > 366.3 *	−21	−20	−25	427.4 > 286.2	−10	−29	−19
QNs	Gatifloxacin	376.3 > 332.4 *	−14	−17	−24	376.3 > 261.4	−14	−29	−17
QNs	Gemifloxacin	390.1 > 372.1 *	−20	−20	−25	390.1 > 313	−21	−29	−21
QNs	Lomefloxacin	352 > 308.1 *	−17	−17	−21	352 > 265	−17	−24	−17
QNs	Marbofloxacin	363.1 > 345.1 *	−18	−20	−24	363.1 > 320	−18	−16	−21
QNs	Moxifloxacin	402.1 > 358.3 *	−12	−21	−25	402.1 > 364.3	−20	−29	−26
QNs	Nadifloxacin	361.3 > 343.3 *	−13	−22	−23	361.3 > 283.3	−13	−38	−19
QNs	Nalidixic acid	233.1 > 215.1 *	−12	−15	−14	233.1 > 187.1	−14	−24	−12
QNs	Norfloxacin	320.1 > 302.1 *	−11	−30	−23	320.2 > 231.1	−16	−46	−28
QNs	Ofloxacin	362.2 > 318.2 *	−12	−19	−22	362.2 > 261.1	−20	−33	−26
QNs	Orbifloxacin	396 > 295 *	−17	−19	−24	396 > 352	−16	−25	−20
QNs	Oxolinic acid	262.1 > 244.1 *	−30	−18	−30	262.1 > 216	−30	−32	−26
QNs	Pefloxacin	334.1 > 316.1 *	−17	−21	−21	334.1 > 290.2	−17	−19	−20
QNs	Pipemidic acid	304.1 > 286.1 *	−15	−19	−19	304.1 > 215.1	−16	−33	−14
QNs	Piromidic acid	289.1 > 271.1 *	−15	−19	−18	289.1 > 243.1	−15	−30	−16
QNs	Rufloxacin	364.3 > 320.1 *	−10	−19	−22	364.3 > 263	−10	−25	−17
QNs	Sitafloxacin	410.1 > 392 *	−20	−20	−27	/	/	/	/
QNs	Sparfloxacin	393.2 > 292 *	−21	−25	−19	393.2 > 349.2	−14	−19	−24
QNs	Tosufloxacin	405.2 > 387.1 *	−14	−23	−27	405.2 > 263.1	−14	−33	−17
QNs	Trovafloxacin	417.3 > 399.4 *	−16	−20	−14	417.3 > 330.2	−12	−32	−22
SAs	Sulfadiazine	251 > 156 *	−12	−15	−15	251 > 92	−12	−24	−16
SAs	Sulfamethazine	279.1 > 186 *	−15	−17	−12	279.1 > 156	−14	−19	−15
SAs	Sulfamethoxazole	281.1 > 156 *	−10	−16	−15	281.1 > 126	−10	−20	−22
SAs	Sulfamethoxypyridazine	254.1 > 156 *	−14	−16	−15	254.1 > 108	−15	−23	−18
SAs	Sulfapyridine	250.1 > 156 *	−14	−16	−15	250.1 > 108	−14	−23	−21
SAs	Sulfaquinoxaline	301.1 > 156 *	−17	−17	−15	301.1 > 108	−18	−26	−10
SAs	Trimethoprim	291.1 > 230.1 *	−20	−22	−25	291.1 > 123.1	−20	−23	−13
MLs	Clarithromycin	748.5 > 158.1 *	−22	−20	−22	748.5 > 590.4	−22	−28	−15
MLs	Erythromycin	734.5 > 576.4 *	−22	−19	−20	734.5 > 158.1	−28	−30	−15
MLs	Roxithromycin	837.5 > 158.1 *	−26	−22	−24	837.5 > 679.4	−24	−35	−15
MLs	Tylosin	916.5 > 174.1 *	−26	−38	−17	916.5 > 772.5	−22	−31	−28
TLs	Chlortetracycline	479.1 > 462 *	−14	−19	−16	479.1 > 444	−14	−20	−15
TLs	Oxytetracycline	461.2 > 426.2 *	−10	−21	−29	461.2 > 443.2	−26	−13	−12
TLs	Tetracycline	445.1 > 410.2 *	−22	−21	−14	445.1 > 427.1	−23	−13	−30

* represents the quantitative ion pair. MRM: Multiple reaction monitoring. CE: collision energy.

**Table 3 foods-13-00031-t003:** Analytical performance of the applied method.

NO.	Compound	Range	LOD (μg/kg)	LOQ (μg/kg)	Equation	R^2^	Rec%
1	Cinoxacin	1–500	0.03	0.1	y = 120,091x + 935,713	0.9973	82.7–91.8
2	Ciprofloxacin	1–500	0.03	0.1	y = 10,623x + 97,493	0.9982	60.7–81.1
3	Danofloxacin	1–500	0.03	0.1	y = 22,557x − 131,399	0.9984	60.7–67.3
4	Difloxacin	1–500	0.03	0.1	y = 2402x + 2859	0.9972	70.4–108
5	Enoxacin	1–500	0.03	0.1	y = 8321x + 8649	0.9991	72.8–105
6	Enrofloxacin	1–500	0.03	0.1	y = 23,695x + 252,731	0.9952	68.7–90.4
7	Fleroxacin	1–500	0.03	0.1	y = 16,727x + 158,880	0.9958	64.2–75.6
8	Flumequine	1–500	0.03	0.1	y = 210,025 x + 2,047,746	0.9941	70.8–88.1
9	Garenoxacin	1–500	0.03	0.1	y = 1223x + 7726	0.9979	71.4–77.7
10	Gatifloxacin	1–500	0.03	0.1	y = 6915x + 17,620	0.9985	66.8–83.9
11	Grepafloxacin	1–500	0.03	0.1	y = 33,297x − 91,235	0.9993	62.7–81.4
12	Lomefloxacin	1–500	0.03	0.1	y = 47,150x + 45,4340	0.9957	60.7–86.1
13	Marbofloxacin	1–500	0.03	0.1	y = 7753x + 62,986	0.9955	70.4–94.7
14	Moxifloxacin	1–500	0.03	0.1	y = 7009x + 59,232	0.9982	66.3–81.2
15	Nadifloxacin	1–500	0.03	0.1	y = 55,559x + 593,107	0.9963	76.9–94.2
16	Nalidixic acid	1–500	0.03	0.1	y = 116,642x + 1,478,565	0.9921	73.4–84.2
17	Norfloxacin	1–500	0.03	0.1	y = 11,599x + 21,913	0.999	66.4–103
18	Ofloxacin	1–500	0.03	0.1	y = 35,951x + 149,646	0.999	63.1–87.3
19	Orbifloxacin	1–500	0.03	0.1	y = 134,217x + 1,462,058	0.992	81.1–88.2
20	Oxolinic acid	1–500	0.03	0.1	y = 958x +35,006	0.9895	76.2–83.3
21	Pefloxacin	1–500	0.03	0.1	y = 37,122x − 51,761	0.9997	64.4–107
22	Piromidic acid	1–500	0.03	0.1	y = 62,512x + 75,905	0.9997	76.1–88.5
23	Pipemidic acid	1–500	0.03	0.1	y = 70,845x + 538,941	0.9978	60.2–86.0
24	Rufloxacin	1–500	0.03	0.1	y = 27,763x + 88,747	0.9994	65.2–73.4
25	Sitafloxacin	1–500	0.03	0.1	y = 13,584x + 14,965	0.9999	85.4–89.2
26	Sparfloxacin	1–500	0.03	0.1	y = 37,891x + 109,631	0.9976	73.9–105
27	Tosufloxacin	1–500	0.03	0.1	y = 7615x + 94,247	0.9988	61.4–90.9
28	Trovafloxacin	1–500	0.03	0.1	y = 14,871x + 34,167	0.9964	93.7–95.9
29	Sulfadiazine	1–500	0.1	0.3	y = 27,637x + 55,614	0.9994	85.1–96.1
30	Sulfamethazine	1–500	0.1	0.3	y = 57,004x + 405,012	0.9956	71.4–80.6
31	Sulfamethoxazole	1–500	0.1	0.3	y = 25,460x + 195,424	0.9965	68.9–99.1
32	Sulfamethoxypyri-dazine	1–500	0.1	0.3	y = 11,574x + 120,293	0.9945	61.4–86.8
33	Sulfapyridine	1–500	0.1	0.3	y = 39,762x + 203,397	0.9981	82.6–98.1
34	Sulfaquinoxaline	1–500	0.1	0.3	y = 19,167x + 309,789	0.9906	77.7–104
35	Trimethoprim	1–500	0.1	0.3	y = 37,479x + 405,194	0.9936	73.8–96.5
36	Clarithromycin	1–500	0.06	0.2	y = 51,246x + 490,620	0.9937	70.7–89.5
37	Erythromycin	1–500	0.06	0.2	y = 8186x + 6662	0.9992	75.1–93.8
38	Roxithromycin	1–500	0.06	0.2	y = 51,084x + 282,466	0.9971	65.0–89.1
39	Tylosin	1–500	0.06	0.2	y = 21,353x + 6637	0.9978	64.6–119
40	Chlortetracycline	1–500	0.1	0.3	y = 7678x + 18,977	0.9946	63.3–84.2
41	Oxytetracycline	1–500	0.1	0.3	y = 12,209x − 48,911	0.9955	68.7–85.8
42	Tetracycline	1–500	0.1	0.3	y = 53,991x − 267,978	0.9915	69.0–85.7

**Table 4 foods-13-00031-t004:** Detection rate and range of antibiotics in the different tissues.

Antibiotics Species	Range (μg/kg)	Detection Rate %								
Quinolones ^a^		muscle	brain	Swim bladder	liver	kidney	intestine	gills	heart	gonad
Ciprofloxacin	0.24–60.58	100	93.33	40	93.33	80	26.67	86.67	80	73.33
Enrofloxacin	1.30–87.22	100	93.33	100	-	73.33	53.33	93.33	-	93.33
Nalidixic acid	ND–0.65	6.67	-	-	-	-	-	-	-	-
Flumequine	0.03–0.60	20	-	-	-	20	-	-	-	-
Oxolinic acid	0.64–2.98	-	-	46.67	-	40	-	-	20	-
Lomefloxacin	0.82–4.23	-	-	13.33	20	-	-	20	-	-
Garenoxacin	9.07–30.29	-	-	-	-	-	-	13.33	-	-
Rufloxacin	1.70–7.74	-	-	-	-	-	-	-	13.33	13.33
Norfloxacin	ND–0.26	-	-	-	-	-	-	-	6.67	-
Pefloxacin	0.15–0.81	-	-	-	-	-	-	-	40	-
Tetracyclines ^a^										
Oxytetracycline	0.30–11.38	20	-	-	-	-	-	20	46.67	-
Chlortetracycline	1.34–5.96	-	13.33	-	-	-	-	-	6.67	-
Macrolides ^a^										
Clarithromycin	0.09–0.66	-	-	-	-	-	-	-	20	13.33
Tylosin	0.13–2.33	-	13.33	-	-	-	-	-	26.67	46.67
Erythromycin	1.13–4.00	-	-	-	-	-	-	-	20	13.33
Sulfonamides ^a^										
Sulfadiazine	0.19–3.92	-	20	-	-	20	53.33	26.67	13.33	-
Trimethoprim	0.21–0.48	-	13.33	-	-	-	20	-	-	-

^a^ sum of antibiotics in corresponding category. ND: Not detected.

**Table 5 foods-13-00031-t005:** Comparison between estimated daily intake (EDI) and acceptable daily intake (ADI).

Area	Target Compound	Concentration (μg/g)	EDI (ng/kg bw/d)	ADI (μg/kg bw/d)	HQ
Site 1	Ciprofloxacin	7.05 × 10^−3^	5.79 × 10^−3^	6.2	9.34 × 10^−4^
	Enrofloxacin	4.65 × 10^−3^	3.82 × 10^−3^	6.2	6.17 × 10^−4^
Site 2	Ciprofloxacin	9.20 × 10^−3^	7.56 × 10^−3^	6.2	1.22 × 10^−4^
	Enrofloxacin	7.07 × 10^−3^	5.81 × 10^−3^	6.2	9.37 × 10^−4^
Site 3	Ciprofloxacin	2.28 × 10^−2^	1.88 × 10^−2^	6.2	3.02 × 10^−3^
	Enrofloxacin	1.55 × 10^−2^	1.27 × 10^−2^	6.2	2.06 × 10^−3^
Site 4	Ciprofloxacin	1.10 × 10^−2^	9.06 × 10^−3^	6.2	1.46 × 10^−3^
	Enrofloxacin	1.08 × 10^−2^	8.86 × 10^−3^	6.2	1.43 × 10^−3^
Site 5	Ciprofloxacin	9.69 × 10^−3^	7.96 × 10^−3^	6.2	1.28 × 10^−3^
	Enrofloxacin	1.81 × 10^−2^	1.49 × 10^−2^	6.2	2.40 × 10^−3^

## Data Availability

The data are available from the corresponding author.

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
