# Peer review of "Health Risk Assessment of Antibiotic Pollutants in Large Yellow Croakers from Zhejiang Aquaculture Sites"

_foods, 2023, doi:10.3390/foods13010031_

Round 1

Reviewer 1 Report

Comments and Suggestions for Authors

Dear Authors,

Thank you for an interesting article with great relevance to the detection of antibiotic residues in human food sources. However, the article contains several issues that should have been addressed.

Introduction

Information on the current situation in the human health risk assessment of antibiotics in fish in the world and in China should be added.

Materials and Methods

- Please, describe why the selected locations were chosen for the fish sampling. Are there any information and analyses done in the selected sites? Are there any data on the antibiotics occurring in the selected sites?

- What were the LOD and LOQ values for analyzed antibiotics?

- What was the time between the catching the fish and the dissection and tissue sampling?

- Table 2 – please, explain abbreviations under the table in the legends

- Pg. 8, row 13 – please, correct The Estimated Daily Fish Intake to Estimated Daily Intake of antibiotics

- What is the source for the estimated daily fish consumption by Chinese adults (49.30 g/day)?

- Why did you express an HI above or under 0.01? Potential health risk is present or absent when the Hazard Index is above/under 1.0.

- Why did you only perform risk assessments for enfofloxacin and ciprofloxacin?

- Which method was used for statistical analysis? What kind of tests?

- The limitation of the study is the low number of samples obtained from each sampling site.

Results

Chapter 3.1 should be part of the methodology.

Conclusion

-         Pg. 18, row 183 - Nevertheless, as there is currently no designated regulatory limit for this tissue.

References

-         Pg. 19, line 238 – reference #18 – missing authors

-         Pg. 20, line 276 – reference #37 does not correspond to the cited reference in the text (pg. 14, line 41-42)

-         Pg. 20, line 288 – reference #43 does not correspond to the cited reference in the text (pg. 16, line 103-105)

-         Pg. 20, line 290 – reference #44 does not correspond to the cited reference in the text (pg. 16, line 105-107)

-         Pg. 21, line 315 – reference #56 is missing in the text

Comments on the Quality of English Language

Minor editing of English language required.

Author Response

Dear Editors and Reviewers,

We would like to thank you for your constructive comments concerning our manuscript (foods-2757616). These comments have helped to further improve the quality of our manuscript and clarify the significance of our research. We have carefully considered all your comments and have made the necessary corrections. Our point-by-point responses to each comment are presented below.

Reviewer: 1

Comments:

Thank you for an interesting article with great relevance to the detection of antibiotic residues in human food sources. However, the article contains several issues that should have been addressed.

Major comments

1.Introduction: Information on the current situation in the human health risk assessment of antibiotics in fish in the world and in China should be added.

Response: Thank you for your suggestion. The modified details you mentioned were presented in “Instruction” parts as follows (Line 87-91): Previous studies have assessed the possible health hazards linked to consuming fish that have been exposed to antibiotics. The consumption of fish in Canada28(Dinh et al,2020) and Egypt29(Eissa et al,2020) demonstrated minimal negative effects on consumers. In contrast, a study in China21(Han et al,2020) found significant risks associated with consuming fish. The references had also been updated and listed as follows:

  1. Han, Q.F.; Zhao, S.; Zhang, X.R.; Wang, X.L.; Song, C.; Wang, S.G. Distribution, Combined Pollution and Risk Assessment of Antibiotics in Typical Marine Aquaculture Farms Surrounding the Yellow Sea, North China. Environ Int. 2020, 138, 105551.
  2. Dinh, Q.T.; Munoz, G.; Vo Duy, S.; Tien Do, D.; Bayen, S.; Sauvé, S. Analysis of Sulfonamides, Fluoroquinolones, Tetracyclines, Triphenylmethane Dyes and Other Veterinary Drug Residues in Cultured and Wild Seafood Sold in Montreal, Canada. J Food Compos Anal. 2020, 94, 103630.
  3. Eissa, F.; Ghanem, K.; Al-Sisi, M. Occurrence and Human Health Risks of Pesticides and Antibiotics in Nile Tilapia along the Rosetta Nile Branch, Egypt. Toxicology Reports. 2020, 7, 1640–1646.
  4. Materials and Methods:

1) Please, describe why the selected locations were chosen for the fish sampling. Are there any information and analyses done in the selected sites? Are there any data on the antibiotics occurring in the selected sites?

Response: We apologize for the lack of clarity. We chose these locations for sampling because aquaculture farms are predominantly located in these coastal regions, and these locations are notably representative of large yellow croaker farming operations within the aquaculture. Our choice of this location based on specific information and a comprehensive analysis. Here are some data of antibiotics occurring in the selected sites. Shao et al. conducted a study on antibiotic usage in areas including Taizhou, Wenzhou, and Ningbo, revealing the presence of quinolone antibiotics in the feed35. Li et al. found that in their study of antibiotic residues in aquaculture regions in eastern China, including Tai Zhou, enrofloxacin was observed detection rate as high as 85%43. Those two sections have been respectively included in the discussions for 3.1 and 3.2 by us.

For better understanding, this part had been revised in Pg.16-Line2-5 and Pg.17Line47-49: For instance, a previous study on antibiotic use in coastal aquaculture within areas like Ningbo, Wenzhou, and Taizhou discovered the presence of quinolone antibiotics incorporated into aquaculture feed, with levels reaching as high as 140 milligrams per kilogram35. (Pg.16-Line2-5) Li et al. found that in their study of antibiotic residues in aquaculture regions in eastern China, including Tai Zhou, enrofloxacin was observed detection rate as high as 85%43. (Pg.17Line47-49) The references had also been updated and listed as follows:

  1. Shao, Y.; Wang, Y.; Yuan, Y.; Xie, Y. A Systematic Review on Antibiotics Misuse in Livestock and Aquaculture and Regulation Implications in China. Science of The Total Environment. 2021, 798, 149205.
  2. Li, J.-Y.; Wen, J.; Chen, Y.; Wang, Q.; Yin, J. Antibiotics in Cultured Freshwater Products in Eastern China: Occurrence, Human Health Risks, Sources, and Bioaccumulation Potential. Chemosphere. 2021, 264, 128441.

2) What were the LOD and LOQ values for analyzed antibiotics?

Response: We apologize for the lack of clarity. The LOD values of QNs, TLs, MLs, and SAs were 0.03 μg/kg, 0.1 μg/kg, 0.06 μg/kg and 0.1 μg/kg, respectively. The LOQ values of QNs, TLs, MLs, and SAs were 0.1 μg/kg, 0.3 μg/kg, 0.2 μg/kg, 0.3 μg/kg, respectively.

For better understanding, this part had been revised in Table 3 and Pg. 9 Line 13-16: The limits of detection (LODs) and limits of quantification (LOQs) were established following the methodology outlined in Shaaban et al.'s research34, which were established using signal-to-noise ratios of 3:1 and 10:1, respectively. The references had also been updated and listed as follows:

  1. Shaaban, H.; Mostafa, A. Simultaneous Determination of Antibiotics Residues in Edible Fish Muscle Using Eco-Friendly SPE-UPLC-MS/MS: Occurrence, Human Dietary Exposure and Health Risk Assessment for Consumer Safety. Toxicology Reports. 2023, 10, 1–10.

3) What was the time between the catching the fish and the dissection and tissue sampling?

Response: We apologize for the lack of clarity. The captured fish samples were swiftly transported to the laboratory within 2 hours while being kept at 4°C during transit. Immediate dissection and sampling were then performed, followed by storage at -20°C.

For better understanding, this part had been revised in Pg. 3 Line 109-110: In each site, three parallel samples were collected and swiftly transported back to the laboratory within 2 hours for immediate dissection and sampling.

4) Table 2 – please, explain abbreviations under the table in the legends

Response: Thank you for your suggestion. We apologize for the lack of clarity. We have explained abbreviations under the table in the legends: MRM: Multiple reaction monitoring; CE: collision energy.

For better understanding, this part had been revised in Table 2.

5) Pg. 8, row 13 – please, correct The Estimated Daily Fish Intake to Estimated Daily Intake of antibiotics.

Response: Thank you for your suggestion. We have already replaced “The Estimated Daily Fish Intake” with “Estimated Daily Intake of antibiotics”.

6) What is the source for the estimated daily fish consumption by Chinese adults (49.30 g/day)?

Response: We apologize for the lack of clarity. The FR (49.30 g/day) data aligns with the information provided in Shen et al.’s research. We have added relevant references. The references had also been updated and listed as follows:

  1. Shen, L.-N.; Fu, Y.; Zhang, L.-L.; Qin, S.; Ju, Z.-J.; Yao, B.; Cui, J.-S. [Bioaccumulation Characteristics of Quinolones (QNs) in Dominant Fish Species and Their Correlation with Environmental Factors in Baiyangdian Lake]. Huan Jing Ke Xue. 2020, 41, 5470–5479.

7) Why did you express an HI above or under 0.01? Potential health risk is present or absent when the Hazard Index is above/under 1.0.

Response: We apologize for the lack of clarity. We have corrected the “HI above or under 0.01” in the article to “HI above or under 1.0”.

8) Why did you only perform risk assessments for enrofloxacin and ciprofloxacin?

Response: We apologize for the lack of clarity. It's crucial to highlight that enrofloxacin and ciprofloxacin were identified as the primary quinolone antibiotics with the highest detection rates and concentrations in this study. The levels of these antibiotics are significantly higher than those of other antibiotics, both in edible and inedible parts. This targeted approach will offer valuable insights into the potential risks associated with these specific antibiotics.

For better understanding, this part had been revised in Pg. 12 Line 51-54: Enrofloxacin and ciprofloxacin are the primary quinolone antibiotics with the highest detection rates and concentrations in this study. Due to their significantly higher detection rates and concentrations compared to other quinolone antibiotics, a dedicated risk assessment was conducted for these specific compounds.

9) Which method was used for statistical analysis? What kind of tests?

Response: We apologize for the lack of clarity. We made an error in that section, and it has been removed. Thank you for your suggestion.

10) The limitation of the study is the low number of samples obtained from each sampling site.

Response: Thank you for your suggestion. We regret the limited sample size in our study, which we have also acknowledged in the conclusion. We will take this into consideration for future research and aim to increase the sample size to enhance the robustness of our findings. Your feedback is greatly appreciated and will be valuable for the improvement of our work.

  1. Results and Discussion:

1) Chapter 3.1 should be part of the methodology.

Response: Thank you for your suggestion. Chapter 3.1 has been relocated to the methodology section.

4.Conclusion

1) Pg. 18, row 183 - Nevertheless, as there is currently no designated regulatory limit for this tissue.

Response: Thank you for your suggestion. We apologize for the lack of clarity. We have already deleted “Nevertheless, as there is currently no designated regulatory limit for this tissue”.

Minor comments

5.References

1) Pg. 19, line 238 – reference #18 – missing authors

Response: We apologize for the lack of clarity. We have now included the relevant authors as per your suggestion. Your input is greatly appreciated! The references had also been updated and listed as follows:

  1. Adenaya, A.; Berger, M.; Brinkhoff, T.; Ribas-Ribas, M.; Wurl, O. Usage of Antibiotics in Aquaculture and the Impact on Coastal Waters. Mar Pollut Bull. 2023, 188, 114645.

2) Pg. 20, line 276 – reference #37 does not correspond to the cited reference in the text (pg. 14, line 41-42)

Response: We apologize for the lack of clarity. We mistakenly listed the author information incorrectly in the article. We have replaced the “Hua” with “Griboff”. (Line 43)

3) Pg. 20, line 288 – reference #43 does not correspond to the cited reference in the text (pg. 16, line 103-105)

Response: We apologize for the lack of clarity. We mistakenly listed the author information incorrectly in the article. We have replaced the “Wang” with “Zhang”. (Line 106)

4) Pg. 20, line 290 – reference #44 does not correspond to the cited reference in the text (pg. 16, line 105-107)

Response: We apologize for the lack of clarity. We mistakenly listed the author information incorrectly in the article. We have replaced the “He” with “Hua”. (Line 109)

5) Pg. 21, line 315 – reference #56 is missing in the text

Response: We apologize for the lack of clarity. Thank you for your suggestion. Following your suggestion, we've identified an inconsistency between our reference and the citation in the text.  We've made the necessary revisions to ensure accuracy. Your feedback was invaluable in improving our work.

Reviewer 2 Report

Comments and Suggestions for Authors

This paper aimed to analyze the distribution of antibiotics in yellow croaker, a common fish in Asian aquaculture facilities with a significant commercial and economic value in China. 

The authors have looked for the possible presence of 42 antibiotics in both edible and inedible parts of the sampled fish using the UPLC-MS/MS technique and confirmed the presence of 17 of these. Quinolones were the most detected antibiotics, especially enrofloxacin and its metabolite ciprofloxacin, suggesting the possible threat to human health due to the consumption of tainted fish.   

However, some issues need to be addressed:

- line 1: please replace “combatting declining” with “combatting the decline of”.

- line 2: please replace “aqua-culture” with “aquaculture”.

- line 17: please replace “quinolone” with “quinolones”.

- line 21: please replace “… muscle, there …” with “… muscle, but there …”.

- line 26: please use correctly the acronyms as they appear in the text and not randomly (“QNs” )

- line 35: SPSS 20.0 was used for statistical 37 analyses…Which ones?

- lines 66-67: please replace the dot with a comma and replace “In the course of this extensive utilization, it is often accompanied by” with “leading to”

- line 71: please replace “undergo” with “undergoes”.

- line 84: please replace “are crucial” with “is crucial” .

- line 87: please replace “43” with “42”.

- line 106: please replace “ten distinct sections” with “nine distinct sections”.

- line 107: please replace “43” with “42”.

- line 109: please rephrase with  “The solvent used were…”.

- Tab. 1: Same as line 26 what are “FQNs”? Fluoroquinolones?

- line 122: please replace “will be” with “were”.

- line 123: please replace “will be” with “were”.

- line 138: please replace “condition” with “conditions”.

- line 142: please reduce the space between the dot and “eluent”.

- Tab. 2: Same as in Tab 1

Starting from page 8, a new line counting begins, so from now on the line numbers are indicated starting from page 8.

- line 7: please replace “43” with “42”.

- line 9: “chosen to spiked” ? Please explain the meaning

- line 52: please replace “conducted” with “evaluated”.

- line 55: please replace “account” with “accounted”.

- line 57: please replace “garenoxacin with the highest levels were observed in fish gills” with “garenoxacin showed its highest concentration in fish gills”.

- line 59: please replace “constitute” with “constituted”.

- line 63: please replace “account for” with “accounted for”.

Starting from page 13, a new line counting begins, so from now on the line numbers are indicated starting from page 13.

- line 3: please replace “… low water solubility tend…” with “low water solubility, and as a consequence tend…”

- line 9: please replace “…own research results, QNs dominate…” with “…own research results, where QNs dominate…”

- line 11: please replace “Our results showed that the frequent detection of 10 QNs were found in large yellow croaker, possibly due to their extensive use in this area…” with “Our results showed that the frequent detection of 10 QNs highlighted in large yellow croaker is possibly due to their extensive use in this area…”

- line 16: please replace “the distribution of detected QNs in were analyzed…” with “the distribution of detected QNs was analyzed…”

- lines 18-20: it is reported that enrofloxacin levels were the second highest levels in gills, but it goes in contrast with the indicated data, in fact garenoxacin is reported to have a concentration of 4.89 μg/kg, while enrofloxacin had a concentration of 8.62 μg/kg. On the other hand, fig. 3 confirms that the concentration of enrofloxacin is lower than the one showed by garenoxacin as the orange corresponding to garenoxacin in gills is darker than the one observed for enrofloxacin, but the orange corresponding to garenoxacin in heart is lighter than the one of garenoxacin in gills, so it is impossible that the garenoxacin concentration in heart is 15.19 μg/kg. Maybe in line 18 it should be replaced “In hearts and gills” with “in gills and heart”?

- 23: please replace “enrofloxacin were” with “enrofloxacin was” and “kidney” with “kidneys”;

- line 26: please replace “intestines” with “intestine”.

- line 27: please remove “with” in “with the highest concentration” and replace “livers” with “liver”.

- line 29: please replace “enrofloxacin were” with “enrofloxacin was”.

- line 32: please remove the “0” in “ciprofloxa0cin”.

- Fig 3: please remove “the” both in the picture and in its caption.

- line 44: please remove “the” in “the highest the residue”.

- lines 52-54: please replace “The occurrence of the ciprofloxacin in these cultured fish can be explained by one reasons: the occurrence of ciprofloxacin in aquatic products is primarily influenced by the degradation of enrofloxacin” with “The occurrence of ciprofloxacin in these cultured fish can be explained by considering that it is primarily influenced by the degradation of enrofloxacin”.

- line 66: please replace “livers” with “liver” and “the larger” with “the largest”.

- line 67: please replace “livers” with “liver”.

- line 70: please replace “kidney” with “kidneys” and “livers” with “liver”.

- line 71: please replace “kidney” with “kidneys” and “livers” with “liver”.

- line 72: please replace “Among 5 coastal aquacultures, the site3 was detected…” with “Among the 5 coastal aquacultures analyzed, in site3 was detected…”.

- line 74: please replace “account” with “accounted” and “the larger” with “the largest”.

- line 79: muscle is repeated twice, maybe one of the two percentages reported is referred to swim bladder.

- line 79-80: please replace “at inedible parts” with “within the inedible parts”.

- lines 85-86: please replace “kidney” with “kidneys”.

Fig 4: please replace “kidney” with “kidneys” and “livers” with “liver” in the picture.

- line 94-95: please replace “converting” with “converting it”.

- line 99: please remove “metabolite”.

- line 110-111: please replace “Although in edible parts the portion of muscle were not the highest except Site1 and Site2…” with “Although within the edible parts the antibiotic concentration in muscle were not the highest ones except in Site1 and Site2…”.

- line 112: please replace “due to that” with “due to the fact that”.

- line 115: where is this limit of 100 μg/kg for muscle reported? Which is the source?

- line 123: please replace “are” with “were”.

- line 124: please replace “calculate” with “calculated”.

- line 136: please replace “for 5 aqua-culture” with “for all the 5 aquaculture areas analyzed”. 

- line 148: please replace “enrofloxacin have been” with “enrofloxacin has been”.

- line 149: please replace “due to the risks they” with “due to the risks it”.

- line 161: please remove “in” in “in previous studies”.

- line 182: please remove “as” in “as there is currently…”.

- References: please replace the comma with a semicolon after the name of each author in all the references. Furthermore, please format the year in bold and put it after the name of each journal, that has to be formatted in italics in reported in its abbreviated form. Please remember that the volume number has to be reported after the year and formatted in italics, followed by the page range.

- References: please replace “&” with a semicolon, there it is present.

- References:  please replace “et. al” with the complete list of authors.

- lines 261-262: please remove “chaoter 4” and “(ed. Hashmi, M.Z.) and format it as previously described.  

Comments on the Quality of English Language

I strongly recommend to improve the English grammar since there are many point within the text that prevent the reading flow.

Author Response

Dear Editors and Reviewers,

We would like to thank you for your constructive comments concerning our manuscript (foods-2757616). These comments have helped to further improve the quality of our manuscript and clarify the significance of our research. We have carefully considered all your comments and have made the necessary corrections. Our point-by-point responses to each comment are presented below.

Reviewer: 2

Comments:

This paper aimed to analyze the distribution of antibiotics in yellow croaker, a common fish in Asian aquaculture facilities with a significant commercial and economic value in China.

The authors have looked for the possible presence of 42 antibiotics in both edible and inedible parts of the sampled fish using the UPLC-MS/MS technique and confirmed the presence of 17 of these. Quinolones were the most detected antibiotics, especially enrofloxacin and its metabolite ciprofloxacin, suggesting the possible threat to human health due to the consumption of tainted fish.

However, some issues need to be addressed:

Major comments

  1. Key word

1) line 26: please use correctly the acronyms as they appear in the text and not randomly (“QNs”)

Response: We apologize for the lack of clarity. We have replaced the “QNs” with “quinolones”.

  1. Materials and Methods:

1) SPSS 20.0 was used for statistical 37 analyses…Which ones?

Response: We apologize for the lack of clarity. We made an error in that section, and it has been removed. Thank you for your suggestion.

2) line 9: “chosen to spiked”? Please explain the meaning

Response: We apologize for any confusion. In this context, we aim to analyze the recovery rate. To achieve this, we have made adjustments to this sentence.

For better understanding, this part had been revised in Pg 8 Line 8-10: Blank samples were spiked with target antibiotics at three concentration levels (1, 10, and 50 μg/kg) with five replicates to confirm recovery percentages.

  1. Results and Discussion:

1) line 115: where is this limit of 100 μg/kg for muscle reported? Which is the source?

Response: We apologize for the lack of clarity. This residue limit refers to regulations within China. The 100 μg/kg limit for muscle tissue was established in accordance with the Chinese Food Safety Standard Maximum Residue Limits for Veterinary Drugs in Foods52.

  1. GB 31650-2019; National Food Safety Standard—Maximum Residue Limits for Veterinary Drugs in Foods. Ministry of Agriculture and Rural Affairs of PRC: Beijing, China, 2019.

Minor comments

1) line 1: please replace “combatting declining” with “combatting the decline of”.

Response: Thank you for your suggestion. We have replaced “combatting declining” with “combatting the decline of”.

2) line 2: please replace “aqua-culture” with “aquaculture”.

Response: Thank you for your suggestion. We have replaced “aqua-culture” with “aquaculture”.

3) line 17: please replace “quinolone” with “quinolones”.

Response: Thank you for your suggestion. We have replaced “quinolone” with “quinolones”.

4) line 21: please replace “… muscle, there …” with “… muscle, but there …”.

Response: Thank you for your suggestion. We have replaced “… muscle, there …” with “… muscle, but there …”.

5) lines 66-67: please replace the dot with a comma and replace “In the course of this extensive utilization, it is often accompanied by” with “leading to”.

Response: Thank you for your suggestion. We have replaced the dot with a comma and have replaced “In the course of this extensive utilization, it is often accompanied by” with “leading to”.

6) line 71: please replace “undergo” with “undergoes”.

Response: Thank you for your suggestion. We have replaced “undergo” with “undergoes”.

7) line 84: please replace “are crucial” with “is crucial”.

Response: Thank you for your suggestion. We have replaced “are crucial” with “is crucial”.

8) line 87: please replace “43” with “42”.

Response: Thank you for your suggestion. We have replaced “43” with “42”.

9) line 106: please replace “ten distinct sections” with “nine distinct sections”.

Response: Thank you for your suggestion. We have replaced “ten distinct sections” with “nine distinct sections”.

10) line 107: please replace “43” with “42”.

Response: Thank you for your suggestion. We have replaced “43” with “42”.

11)  line 109: please rephrase with “The solvent used were…”.

Response: Thank you for your suggestion. We have rephrased “The solvent used were methanol of HPLC gradient grade and procured from Merck KGaA in Darmstadt, Germany”.

12) Tab. 1: Same as line 26 what are “FQNs”? Fluoroquinolones?

Response: We apologize for the lack of clarity. We have replaced “FQNs” with “QNs”.

13) line 122: please replace “will be” with “were”. line 123: please replace “will be” with “were”.

Response: Thank you for your suggestion. We have replaced “will be” with “were”.

14) line 138: please replace “condition” with “conditions”.

Response: Thank you for your suggestion. We have replaced “condition” with “conditions”.

15) line 142: please reduce the space between the dot and “eluent”.

Response: Thank you for your suggestion. We have reduced the space between the dot and “eluent”.

16) Tab. 2: Same as in Tab 1

Response: We apologize for the lack of clarity. We have replaced “FQNs” with “QNs”.

17) Starting from page 8, a new line counting begins, so from now on the line numbers are indicated starting from page 8. line 7: please replace “43” with “42”.

Response: We apologize for the lack of clarity. We have replaced “43” with “42”.

18) Tab. 2: Same as in Tab 1.

Response: We apologize for the lack of clarity. We have replaced “FQNs” with “QNs”.

19) line 52: please replace “conducted” with “evaluated”.

Response: We apologize for the lack of clarity. We have replaced “conducted” with “evaluated”.

20) line 55: please replace “account” with “accounted”.

Response: We apologize for the lack of clarity. We have replaced “account” with “accounted”.

21) line 57: please replace “garenoxacin with the highest levels were observed in fish gills” with “garenoxacin showed its highest concentration in fish gills”.

Response: We apologize for the lack of clarity. We have replaced “garenoxacin with the highest levels were observed in fish gills” with “garenoxacin showed its highest concentration in fish gills”.

22) line 59: please replace “constitute” with “constituted”.

Response: We apologize for the lack of clarity. We have replaced “constitute” with “constituted”.

23) line 63: please replace “account for” with “accounted for”.

Response: We apologize for the lack of clarity. We have replaced “account for” with “accounted for”.

24) Starting from page 13, a new line counting begins, so from now on the line numbers are indicated starting from page 13. line 3: please replace “… low water solubility tend…” with “low water solubility, and as a consequence tend…”.

Response: We apologize for the lack of clarity. We have replaced “… low water solubility tend…” with “low water solubility, and as a consequence tend…”.

25)  line 9: please replace “…own research results, QNs dominate…” with “…own research results, where QNs dominate…”.

Response: We apologize for the lack of clarity. We have replaced “…own research results, QNs dominate…” with “…own research results, where QNs dominate…”.

26) line 11: please replace “Our results showed that the frequent detection of 10 QNs were found in large yellow croaker, possibly due to their extensive use in this area…” with “Our results showed that the frequent detection of 10 QNs highlighted in large yellow croaker is possibly due to their extensive use in this area…”.

Response: We apologize for the lack of clarity. We have replaced “Our results showed that the frequent detection of 10 QNs were found in large yellow croaker, possibly due to their extensive use in this area…” with “Our results showed that the frequent detection of 10 QNs highlighted in large yellow croaker is possibly due to their extensive use in this area…”.

27) line 16: please replace “the distribution of detected QNs in were analyzed…” with “the distribution of detected QNs was analyzed…”.

Response: We apologize for the lack of clarity. We have replaced “the distribution of detected QNs in were analyzed…” with “the distribution of detected QNs was analyzed…”.

28) lines 18-20: it is reported that enrofloxacin levels were the second highest levels in gills, but it goes in contrast with the indicated data, in fact garenoxacin is reported to have a concentration of 4.89 μg/kg, while enrofloxacin had a concentration of 8.62 μg/kg. On the other hand, fig. 3 confirms that the concentration of enrofloxacin is lower than the one showed by garenoxacin as the orange corresponding to garenoxacin in gills is darker than the one observed for enrofloxacin, but the orange corresponding to garenoxacin in heart is lighter than the one of garenoxacin in gills, so it is impossible that the garenoxacin concentration in heart is 15.19 μg/kg. Maybe in line 18 it should be replaced “In hearts and gills” with “in gills and heart”?

Response: We apologize for the lack of clarity. We have replaced “In hearts and gills” with “in gills and heart”.

29) 23: please replace “enrofloxacin were” with “enrofloxacin was” and “kidney” with “kidneys”;

Response: We apologize for the lack of clarity. We have replaced “enrofloxacin were” with “enrofloxacin was” and “kidney” with “kidneys”.

30) line 26: please replace “intestines” with “intestine”.

Response: We apologize for the lack of clarity. We have replaced “intestines” with “intestine”.

31) line 27: please remove “with” in “with the highest concentration” and replace “livers” with “liver”.

Response: We apologize for the lack of clarity. We have replaced “livers” with “liver” and removed “with” in “with the highest concentration”.

32) line 29: please replace “enrofloxacin were” with “enrofloxacin was”.

Response: We apologize for the lack of clarity. We have replaced “enrofloxacin were” with “enrofloxacin was”.

33) line 32: please remove the “0” in “ciprofloxa0cin”.

Response: We apologize for the lack of clarity. We have removed the “0” in “ciprofloxa0cin”.

34) Fig 3: please remove “the” both in the picture and in its caption.

Response: We apologize for the lack of clarity. We have removed “the” both in the picture and in its caption.

35) line 44: please remove “the” in “the highest the residue”.

Response: We apologize for the lack of clarity. We have removed “the” in “the highest the residue”.

36) lines 52-54: please replace “The occurrence of the ciprofloxacin in these cultured fish can be explained by one reasons: the occurrence of ciprofloxacin in aquatic products is primarily influenced by the degradation of enrofloxacin” with “The occurrence of ciprofloxacin in these cultured fish can be explained by considering that it is primarily influenced by the degradation of enrofloxacin”.

Response: We apologize for the lack of clarity. We have replaced “The occurrence of the ciprofloxacin in these cultured fish can be explained by one reasons: the occurrence of ciprofloxacin in aquatic products is primarily influenced by the degradation of enrofloxacin” with “The occurrence of ciprofloxacin in these cultured fish can be explained by considering that it is primarily influenced by the degradation of enrofloxacin”.

37) line 66: please replace “livers” with “liver” and “the larger” with “the largest”.

Response: We apologize for the lack of clarity. We have replaced “livers” with “liver” and “the larger” with “the largest”.

38) line 67: please replace “livers” with “liver”.

Response: We apologize for the lack of clarity. We have replaced “livers” with “liver”.

39) line 70: please replace “kidney” with “kidneys” and “livers” with “liver”.

Response: We apologize for the lack of clarity. We have replaced “kidney” with “kidneys” and “livers” with “liver”.

40) line 71: please replace “kidney” with “kidneys” and “livers” with “liver”.

Response: We apologize for the lack of clarity. We have replaced “kidney” with “kidneys” and “livers” with “liver”.

41) line 72: please replace “Among 5 coastal aquacultures, the site3 was detected…” with “Among the 5 coastal aquacultures analyzed, in site3 was detected…”.

Response: We apologize for the lack of clarity. We have replaced “Among 5 coastal aquacultures, the site3 was detected…” with “Among the 5 coastal aquacultures analyzed, in site3 was detected…”.

42) line 74: please replace “account” with “accounted” and “the larger” with “the largest”.

Response: We apologize for the lack of clarity. We have replaced “account” with “accounted” and “the larger” with “the largest”.

43) line 79: muscle is repeated twice, maybe one of the two percentages reported is referred to swim bladder.

Response: We apologize for the lack of clarity. We have replaced one of the “muscle” with “swim bladder.”.

44) line 79-80: please replace “at inedible parts” with “within the inedible parts”.

Response: We apologize for the lack of clarity. We have replaced “at inedible parts” with “within the inedible parts”.

45) lines 85-86: please replace “kidney” with “kidneys”.

Response: We apologize for the lack of clarity. We have replaced “kidney” with “kidneys”.

46) Fig 4: please replace “kidney” with “kidneys” and “livers” with “liver” in the picture.

Response: We apologize for the lack of clarity. We have replaced “kidney” with “kidneys” and “livers” with “liver” in the picture.

47) line 94-95: please replace “converting” with “converting it”.

Response: We apologize for the lack of clarity. We have replaced “converting” with “converting it”.

48) line 99: please remove “metabolite”.

Response: We apologize for the lack of clarity. We have removed “metabolite”.

49) line 110-111: please replace “Although in edible parts the portion of muscle were not the highest except Site1 and Site2…” with “Although within the edible parts the antibiotic concentration in muscle were not the highest ones except in Site1 and Site2…”.

Response: We apologize for the lack of clarity. We have replaced “Although in edible parts the portion of muscle were not the highest except Site1 and Site2…” with “Although within the edible parts the antibiotic concentration in muscle were not the highest ones except in Site1 and Site2…”.

50) line 112: please replace “due to that” with “due to the fact that”.

Response: We apologize for the lack of clarity. We have replaced “due to that” with “due to the fact that”.

51) line 123: please replace “are” with “were”.

Response: We apologize for the lack of clarity. We have replaced “are” with “were”.

52) line 124: please replace “calculate” with “calculated”.

Response: We apologize for the lack of clarity. We have replaced “calculate” with “calculated”.

53) line 136: please replace “for 5 aqua-culture” with “for all the 5 aquaculture areas analyzed”.

Response: We apologize for the lack of clarity. We have replaced “for 5 aqua-culture” with “for all the 5 aquaculture areas analyzed”.

54) line 148: please replace “enrofloxacin have been” with “enrofloxacin has been”.

Response: We apologize for the lack of clarity. We have replaced “enrofloxacin have been” with “enrofloxacin has been”.

55) line 149: please replace “due to the risks they” with “due to the risks it”.

Response: We apologize for the lack of clarity. We have replaced “due to the risks they” with “due to the risks it”.

56) line 161: please remove “in” in “in previous studies”.

Response: We apologize for the lack of clarity. We have removed “in” in “in previous studies”.

57) line 182: please remove “as” in “as there is currently…”.

Response: We apologize for the lack of clarity. We have removed “as” in “as there is currently…”.

58) References: please replace the comma with a semicolon after the name of each author in all the references. Furthermore, please format the year in bold and put it after the name of each journal, that has to be formatted in italics in reported in its abbreviated form. Please remember that the volume number has to be reported after the year and formatted in italics, followed by the page range.

Response: Thank you for your instructions. We have adjusted the format of all the references according to your requirements.

60) References: please replace “&” with a semicolon, there it is present.

Response: Thank you for your instructions. We have replaced “&” with a semicolon.

61) References: please replace “et. al” with the complete list of authors.

Response: Thank you for your instructions. We have replaced “et. al” with the complete list of authors in all references.

62) lines 261-262: please remove “chaoter 4” and “(ed. Hashmi, M.Z.) and format it as previously described.

Response: Thank you for your instructions. We have removed “chaoter 4” and “(ed. Hashmi, M.Z.). The references had also been updated and listed as follows:

  1. Ibrahim, M.; Ahmad, F.; Yaqub, B.; Ramzan, A.; Imran, A.; Afzaal, M.; Mirza, S.A.; Mazhar, I.; Younus, M.; Akram, Q.; et al. Current Trends of Antimicrobials Used in Food Animals and Aquaculture. In Antibiotics and Antimicrobial Resistance Genes in the Environment; Advances in Environmental Pollution Research series; Elsevier, 2020; Vol. 1, pp. 39–69 ISBN 978-0-12-818882-8.

Round 2

Reviewer 2 Report

Comments and Suggestions for Authors

The authors addressed all comments. The paper can now be published